# A Unified View of Evaluation Metrics for Structured Prediction

**Yunmo Chen**[1*]    **William Gantt**[2*]    **Tongfei Chen**[3*]
**Aaron Steven White**[2]    **Benjamin Van Durme**[1]
[1] Johns Hopkins University    [2] University of Rochester    [3] Microsoft

{yunmo|vandurme}@jhu.edu, {wgantt@ur.|aaron.white@}rochester.edu, tongfei@pm.me

## Abstract

We present a conceptual framework that unifies a variety of evaluation metrics for different structured prediction tasks (e.g. event and relation extraction, syntactic and semantic parsing). Our framework requires representing the outputs of these tasks as objects of certain data types, and derives metrics through *matching of common substructures*, possibly followed by *normalization*. We demonstrate how commonly used metrics for a number of tasks can be succinctly expressed by this framework, and show that new metrics can be naturally derived in a bottom-up way based on an output structure. We release a library that enables this derivation to create new metrics.[1] Finally, we consider how specific characteristics of tasks motivate metric design decisions, and suggest possible modifications to existing metrics in line with those motivations.

## 1 Introduction

A wide range of tasks in NLP can be considered as forms of *structured prediction*. Syntactic and semantic parsing produces a tree or graph[2] based on text. Information extraction (IE) aims to produce structured representations of data extracted from unstructured sources, often in the form of relations that may be used to populate a database (Grishman, 2019). Such relations may be typed or untyped, may have different numbers of arguments, and may relate objects of different kinds (e.g. mentions, entities, events, or even images).

The structural complexity of these representations varies considerably between tasks. On the simpler end, problems like *binary relation extraction* require identifying relationships between pairs of entity mentions. On the more complex end are

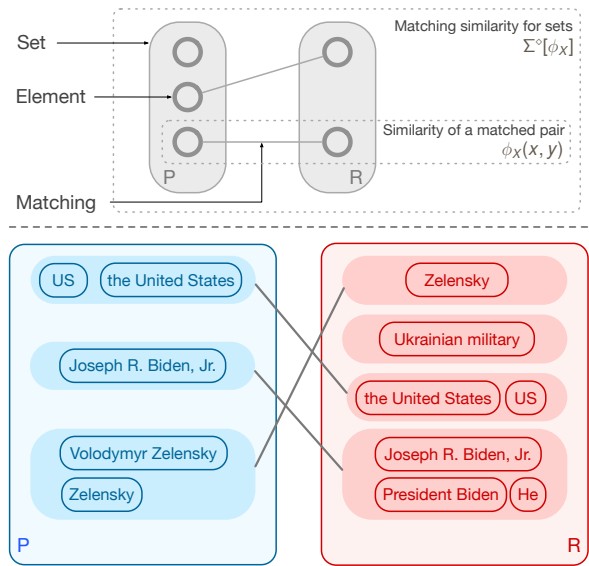

Figure 1: Our generic framework, with the CEAF$_{\phi_4}$ metric (Luo, 2005) as for coreference resolution as an example. Here the task output is a set of entities, where each entity is a set of coreferent mentions identfied in the document. Computing CEAF$_{\phi_4}$ thus amounts to calculating the matching similarity between the predicted ($P$) and reference ($R$) sets of entities.

tasks like *template extraction*, which requires populating various types of slots with *sets* of mentions, categorical values, or even whole event structures, and *AMR parsing* (Langkilde and Knight, 1998; Banarescu et al., 2013), which requires generating a DAG of entities and values representing their semantic relations.

A wide array of evaluation metrics have been proposed across this spectrum of tasks. For simpler ones, researchers have generally converged to a standardized set of metrics (e.g. trigger and argument F$_1$ for event extraction). However, for more complex tasks like template extraction, researchers have often proposed bespoke metrics tailored to the problem at hand, complicating comparison with prior work on similar problems (Chinchor, 1991, 1992; Du et al., 2021b; Chen et al., 2023).

Given the common goal of predicting structured

---

* Equal contribution.
[1] https://github.com/wanmok/metametric.

[2] Often in the form of a *directed acyclic graph* (DAG), as in the task of AMR parsing.

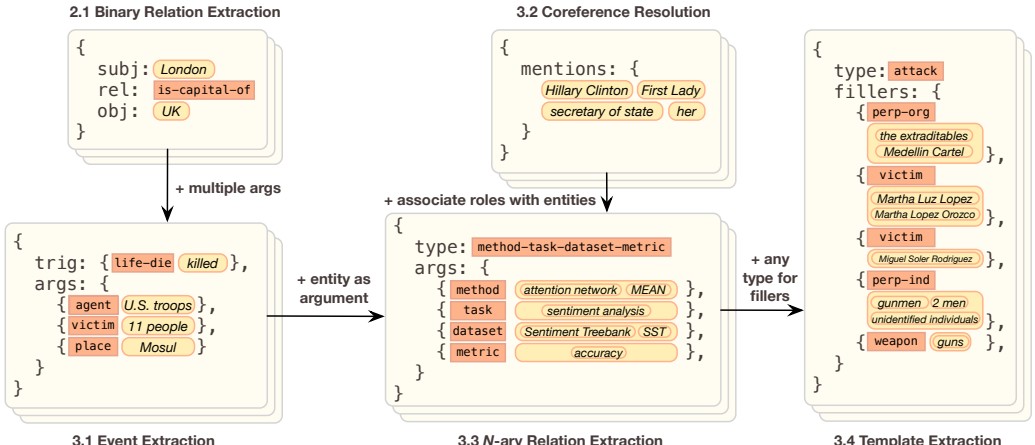

Figure 2: Output structure of common IE tasks discussed in this paper, with examples of their outputs.

objects, our aim is to present a similarly unified, high-level picture of evaluation. We observe that a variety of metrics can be viewed as computing scores over a **matching between substructures** of predicted and reference objects, where this score decomposes as a *normalized sum over matched pairs*. The process of computing metrics can thus be abstracted to a framework as shown in Figure 1.

On the one hand, this observation drives a contribution to structured prediction *theory*, clarifying the relationships among numerous metrics proposed over the years by identifying their core components. On the other, it drives a contribution to NLP *practice*, offering a bottom-up process for designing *new* metrics based on an output structure. Our contributions can be summarized as follows:

- We present a *unified framework* for expressing structured prediction metrics;

- We demonstrate how to derive various classic metrics using this framework, given a specification of a task's output structure;

- We consider how different problem features may recommend particular design decisions within the framework — often different decisions from those realized by existing metrics;

- We release a library that enables bottom-up creation of new metrics based on predefined output data structure of a given task.

Throughout, we emphasize both how evaluation of substructures (e.g. mentions) composes in the evaluation of superstructures (e.g. relations, templates), as well as the different notions of similarity employed for different structures. Our discussion starts with simpler tasks and proceeds to more complex ones, interleaving with examples throughout our exposition.

## 2 Records and Sets

We begin by focusing on records[3] with non-nested, fixed-named *fields* or *slots*. Specifically, for $P, R \in X$ of *Predicted* and *Reference* objects of record type $X$, we induce a *similarity function* over $X$.

**Definition 1.** *A **similarity** over $X$ is a function $\phi \colon X \times X \to [0, 1]$ such that $\forall x, y \in X$, $\phi(x, y) \leq \phi(x, x) = 1$, i.e. an object is at least as similar to itself as to any other. A relaxed version is an **unnormalized** similarity, where $\phi \colon X \times X \to [0, +\infty)$.*

**Discrete Similarity**[4] Equality is a trivial but important notion of similarity, which can be expressed by the Kronecker delta or the Iverson bracket[5] as

$$\delta_X(x, y) = [\![x = y]\!] = \begin{cases} 1 & \text{if } x = y; \\ 0 & \text{if } x \neq y. \end{cases} \quad (1)$$

**Product Similarity for Records** Given two similarities $\phi$ and $\psi$ over sets $X$ and $Y$, we can define a **product similarity** $\phi \times \psi$ for tuples of $X \times Y$:

$$(\phi \times \psi)\,((x, y), (x', y')) = \phi(x, x') \cdot \psi(y, y') \quad (2)$$

Clearly, the product similarity of two similarities is also a similarity.[6] This generalizes to *n*-tuples,

---

[3] In the context of relational databases, a *record* is a *row* (also *named tuple*) describing structured data in a table.

[4] Akin to *discrete metric* and *discrete topology*. Throughout this work we use the word *metric* as it's commonly used in NLP literature: a score for evaluation purposes, rather than the formal mathematical notion that generalizes *distances*.

[5] The Iverson bracket $[\![p]\!]$ is 1 if $p$ is true; otherwise 0.

[6] If at least one similarity in the product is unnormalized, the result is also unnormalized.

or record/class types[7] if a similarity function is defined for each field in the record.

**Set Intersection and Normalization**   Sets are commonly compared with Jaccard similarity, or $F_1$ score. Note that the core of such comparison is the **overlap** between two sets $P, R \subseteq X$, namely

$$\Sigma_\delta(P, R) = |P \cap R| \qquad (3)$$

if we consider the elements of $X$ as discrete (using $\delta_X$ as their similarity). This overlap score $\Sigma_\delta$ is an *unnormalized* similarity under our definition.

There are multiple ways to *normalize* this $\Sigma$ score so that the result is a (proper) similarity. We consider a few common choices: precision (Eq. 4), recall (Eq. 5), and $F_1$ (or *Dice score*; Eq. 6):

$$p = \mathsf{P}(P, R) \;\; = \frac{|P \cap R|}{|P|} \;\; = \frac{\Sigma(P, R)}{\Sigma(P, P)}; \quad (4)$$

$$r = \mathsf{R}(P, R) \;\; = \frac{|P \cap R|}{|R|} \;\; = \frac{\Sigma(P, R)}{\Sigma(R, R)}; \quad (5)$$

$$\mathsf{F}(P, R) \;\; = \frac{2pr}{p + r}; \quad (6)$$

And the Jaccard similarity:

$$\mathsf{J}(P, R) = \frac{|P \cap R|}{|P \cup R|}$$
$$= \frac{\Sigma(P, R)}{\Sigma(P, P) + \Sigma(R, R) - \Sigma(P, R)}. \quad (7)$$

Note that all these normalizers can be expressed solely with the *overlap scoring function* $\Sigma$. Let $\mathsf{N} \in \{\mathsf{P}, \mathsf{R}, \mathsf{F}, \mathsf{J}\}$ be a normalizer over objects of type $X$. Hence we arrive at a normalized similarity over sets of $X$: $\mathsf{N}[\delta](P, R) = \mathsf{N}(\Sigma_\delta(P, R))$.

We have created the basic tools needed to derive metrics for many simple tasks. Next, we illustrate how to do so for two common NLP tasks.

## 2.1   Binary Relation Extraction

Binary relation extraction (RE) focuses on typed relations (e.g. IS-CAPITAL-OF) with two arguments, a *subject* and an *object*. Traditionally, both the subject and the object are text spans (i.e. *mentions*). Given a text passage, the objective is to output a set of binary relations.

To ground our discussion of concrete structured prediction tasks, we specify relevant output data structure(s) in a Python dataclass-like syntax. For binary RE, these are as follows:

---

[7] *Product types* in programming languages literature.

```
class Mention:
  left: int   # left span offset (inclusive)
  right: int  # right span offset (inclusive)

class Relation:
  type: RelationType  # is-capital-of
  subj: Mention       # London
  obj: Mention        # United Kingdom

class RelationSet:    # task output
  relations: Set[Relation]
```

We will now derive a metric bottom-up. A standard similarity for mentions is exact offset match[8], where two mentions are considered the same if and only if both the left and right boundaries match. This is an instance of product similarity:[9]

$$\phi_{\mathtt{Mention}} = \delta_{\mathtt{left}} \times \delta_{\mathtt{right}} \qquad (8)$$

On the outer level, relation instances are considered correct only when all of its components are correct:

$$\phi_{\mathtt{Relation}} = \delta_{\mathtt{type}} \times \delta_{\mathtt{subj}} \times \delta_{\mathtt{obj}} \qquad (9)$$

Finally, precision, recall, and $F_1$ score are the most common metrics to evaluate predicted relations. Practically, this requires finding the intersection between predicted and reference relations:[10]

$$\mathrm{RelF}_1 = \phi_{\mathtt{RelationSet}} = \mathsf{F}_{\mathtt{relations}}[\phi_{\mathtt{Relation}}] \quad (10)$$

## 2.2   Dependency Parsing

Our next example is dependency parsing, where dependencies are relations between a *governor* and its *dependent*. The output structure is as follows:

```
class Dependency:
  gov: int
  dep: int  # index of the word
  rel: DependencyType  # nsubj, advmod, ...

class DependencyParse:  # task output
  edges: Set[Dependency]
```

Dependency parsing is evaluated using unlabeled (UAS) and labeled (LAS) attachment scores (Buchholz and Marsi, 2006), which are simply $F_1$ scores over dependency edges:

$$\mathrm{UAS} = \mathsf{F}_{\mathtt{edges}}\left[\delta_{\mathtt{gov}} \times \delta_{\mathtt{dep}}\right] \qquad (11)$$

$$\mathrm{LAS} = \mathsf{F}_{\mathtt{edges}}\left[\delta_{\mathtt{gov}} \times \delta_{\mathtt{dep}} \times \delta_{\mathtt{rel}}\right] \qquad (12)$$

---

[8] For consistent presentation, we define $\phi_{\mathtt{Mention}}$ in terms of offsets, but other string similarities could be substituted w.l.o.g., e.g. based on string value of the tokens (e.g. bag-of-tokens $F_1$ employed in MRC/QA (Rajpurkar et al., 2016)).

[9] Given a class Cls with fld as a field, we write $\phi_{\mathtt{Cls.fld}}(x, y)$ (or $\phi_{\mathtt{fld}}$ if it is not ambiguous) where $x, y \in$ Cls to mean $\phi_{\mathtt{Cls.fld}}(x, y) = \phi(x.\mathtt{fld}, y.\mathtt{fld})$.

[10] For concision, we present only F in our metric definitions, but precision and recall are defined analogously, substituting P or R for F as appropriate.

## 3 Matching of Sets

In the previous section, we derived $\Sigma_\delta$, a similarity for sets whose elements are *discrete*. However, elements of sets can be equipped with their own similarity. For example, in coreference resolution, the output of a system is a *set* of *entities*, where each entity is in turn a *set* of *mentions* that may *partially* overlap. We develop the idea of a *matching of sets* to express these cases.

We derive a similarity $\phi_{\mathcal{P}(X)}$ over *sets* of elements of $X$ (i.e. elements of the power set $\mathcal{P}(X)$) using *bipartite graphs*. Assuming that elements in $X$ are compared with a custom similarity $\phi_X$, given two sets $P, R \subseteq X$, we can construct a bipartite similarity graph $G = (P, R, E)$ between $P$ and $R$, where $E \subseteq P \times R$ is the set of edges, and the weight on each edge $\phi_X(u, v)$ corresponds to the value of the similarity ($\phi_X$) between nodes $u$ and $v$.

We then determine a *matching* $M^\diamond \subseteq E$ on this bipartite graph $G$. An unnormalized **matching score** between $P$ and $R$ is defined to be the maximum sum of weights of all edges in a matching, subject to some constraint:

$$\Sigma^\diamond[\phi_X](P, R) = \max_{M^\diamond} \sum_{(u,v) \in M^\diamond} \phi_X(u, v), \quad (13)$$

where $\diamond \in \{\leftrightarrow, \rightarrow, \leftarrow, \sim\}$ is the **matching constraint**. Specifically we have the following:

- **1:1 ($\leftrightarrow$):** Each element of $P$ can be matched to at most one element of $R$, and vice versa. This is corresponds to the *unbalanced assignment problem*, and can be solved efficiently with the Hungarian algorithm (Kuhn, 1955; Munkres, 1957). We denote this $M^\leftrightarrow$ since the matching is a (partial) bijection between $P$ and $R$.

- **N:1 ($\rightarrow$) / 1:N ($\leftarrow$):** Each element of $P$ can be matched to at most one element of $R$, but each element of $R$ can be matched to multiple elements of $P$. We denote this $M^\rightarrow$ since the matching is a (partial) function from $P$ to $R$. A flipped version $M^\leftarrow$ obviously follows.

- **N:N ($\sim$):** Every element of $P$ may be matched with multiple elements of $R$, and vice versa, without constraints. We denote this $M^\sim = E$, as the matching may be any relation between $P$ and $R$.

Note that the *overlap score* developed in §2 is a special case of the 1:1 *matching score* here, since

$$\Sigma_\delta(P, R) = |P \cap R| = \Sigma^\leftrightarrow[\delta](P, R). \quad (14)$$

Thus we arrived at a generalization of our original overlap score. We denote the N-normalized ($\mathsf{N} \in \{\mathsf{P}, \mathsf{R}, \mathsf{F}, \mathsf{J}\}$) matching score $\Sigma^\diamond[\phi_X]$ simply as $\mathsf{N}^\leftrightarrow[\phi_X]$. Such (normalized) matching scores are sometimes *kernels*, which have additional nice properties. For discussion, see Appendix B.

With the notion of matching of sets, we next consider metrics for several more complex tasks.

### 3.1 Event Extraction

Our first such task is event extraction.[11] We imagine that events and arguments are represented using the following data structures:

```
class Trigger:
  mention: Mention   # defined in §2.1
  type: EventType

class Argument:
  mention: Mention
  role: RoleType

class Event:
  trig: Trigger
  args: Set[Argument]

class EventSet:   # task output
  events: Set[Event]
```

The canonical metrics for event extraction are labeled precision, recall, and $F_1$ score for both event triggers and arguments (Li et al., 2013, *i.a.*). An event trigger is considered correct iff both the event type and the trigger mention offsets exactly match those of the reference (i.e. $\delta_{\mathsf{Trigger}} = \delta_{\mathsf{mention}} \times \delta_{\mathsf{type}}$). An event argument is considered correct iff the argument mention offsets and role exactly match the reference (i.e. $\delta_{\mathsf{Argument}} = \delta_{\mathsf{mention}} \times \delta_{\mathsf{role}}$) *and* the associated trigger is correct.[12] Given these, we can express trigger and argument $F_1$ scores as

$$\mathrm{TrigF}_1 = \mathsf{F}^\leftrightarrow_{\mathsf{events}}[\delta_{\mathsf{trig}}]; \quad (15)$$

$$\mathrm{ArgF}_1 = \mathsf{F}^\leftrightarrow_{\mathsf{events}}\left[\delta_{\mathsf{trig}} \times \Sigma^\leftrightarrow_{\mathsf{args}}[\delta_{\mathsf{Argument}}]\right]. \quad (16)$$

Note that the definition of $\mathrm{ArgF}_1$ suggests that the metric can be viewed as a *nested* matching, in which we first compute an *unnormalized* optimal argument matching score ($\Sigma^\leftrightarrow_{\mathsf{args}}$, i.e., a raw count of matched arguments) based only on role type and argument boundaries, and then use this score to identify the optimal matching and score conditioned on

---

[11] This also covers *semantic role labeling* (SRL), which is evaluated in the same way, and *event argument extraction* (EAE), which differs only in considering arguments occurring outside the sentence containing the trigger.

[12] Unlabeled scores, in which event type and argument role are ignored, are also commonly reported.

the trigger. As with $F^{\leftrightarrow}_{\text{relations}}$ in §2.1, $\delta_{\text{trig}}$ renders $F^{\leftrightarrow}_{\text{events}}$ trivial to compute, as an aligned event pair receives no credit if the triggers do not match. However, this nested matching view articulates a key aspect of our framework, evidenced by other metrics discussed in this section — namely, that *evaluation of complex structures depends on an optimal matching of their substructures.*

## 3.2 Coreference Resolution

Event extraction deals only with trigger and argument *mentions*, but IE also deals with coreference resolution, where systems predict a *set* of entities, which in turn are *sets* of coreferent mentions:[13]

```
class Entity:
  mentions: Set[Mention]

class EntitySet:   # task output
  entities: Set[Entity]
```

A variety of metrics have been proposed for coreference resolution. Commonly used are CEAF (Luo, 2005), MUC (Vilain et al., 1995) and $B^3$ (Bagga and Baldwin, 1998).

**CEAF** We start with CEAF since it explicitly evaluates coreferences as sets of mentions. CEAF computes entity precision, recall, and $F_1$ by finding a partial bijection between predicted and reference entities that maximizes an entity similarity. Luo (2005) considers several functions – denoted $\phi_{\{1,2,3,4\}}$ – ultimately preferring $\phi_3$ and $\phi_4$:

$$\phi_3 = \Sigma^{\leftrightarrow}_{\text{mentions}} [\delta_{\text{Mention}}] ; \quad (17)$$

$$\phi_4 = F^{\leftrightarrow}_{\text{mentions}} [\delta_{\text{Mention}}] ; \quad (18)$$

Both correspond to intuitive notions of entity similarity, with $\phi_3$ simply counting the number of mentions a pair of entities have in common, while $\phi_4$ F-normalizes this value.[14] In contrast to the identity similarities ($\delta$'s) typically used for mentions, the similarity used in coreference resolution is *gradient*: entities can be more or less correct based on their constituent mentions. Coreference resolution researchers have often used $\phi_4$ (Moosavi and Strube, 2016; Joshi et al., 2020, *i.a.*), where CEAF$_{\phi_4}$ is just the F-normalized total score under a $\phi_4$-optimal entity matching:

$$\text{CEAF}_{\phi_4} = F^{\leftrightarrow}_{\text{entities}} [\phi_4] \quad (19)$$
$$= F^{\leftrightarrow}_{\text{entities}} \left[ F^{\leftrightarrow}_{\text{mentions}} [\delta_{\text{Mention}}] \right] .$$

---

[13] We focus on entity coreference here, though the same metrics can be used for event coreference.

[14] Note that $\phi_3$ is an *unnormalized* similarity function.

CEAF offers a nice illustration of the expressiveness of our framework, computing a matching score between sets (of entities), where the internal metric over elements (entities) is *also* a matching score over sets (of mentions).

**MUC** The main step of MUC scoring is to create (separate) partitions of the predicted and reference entities (Pradhan et al., 2014). Assume that the predicted and reference entity sets are $\mathcal{P}$ and $\mathcal{R}$, and the *partition* of each reference entity $R \in \mathcal{R}$ created by intersecting it with predicted entities $\mathcal{P}$ is $\text{Part}_{\mathcal{P}}(R)$: i.e. $\bigcup_{I \in \text{Part}_{\mathcal{P}}(R)} = R$. MUC recall is computed as

$$R_{\text{MUC}} = \frac{\sum_{R \in \mathcal{R}} (|R| - |\text{Part}_{\mathcal{P}}(R)|)}{\sum_{R \in \mathcal{R}} (|R| - 1)} . \quad (20)$$

Note that $|R| - |\text{Part}_{\mathcal{P}}(R)| = \sum_{I \in \text{Part}_{\mathcal{P}}(R)} (|I| - 1)$: We can define an unnormalized similarity (number of shared links that link mentions to form a coreference chain) between entities:

$$\phi_{\text{link}}(X, Y) = \max\{0, |X \cap Y| - 1\}. \quad (21)$$

Using this, we see that $|R| - |\text{Part}_{\mathcal{P}}(R)| = \Sigma^{\sim}_{\text{entities}} [\phi_{\text{link}}](\mathcal{P}, \mathcal{R})$ with the $N{:}N$ ($\sim$) matching constraint, and $|\text{Part}_{\mathcal{R}}(R)| = 1$. Thus we have

$$R_{\text{MUC}} = R^{\sim}_{\text{entities}} [\phi_{\text{link}}]. \quad (22)$$

Precision can be defined similarly.

**$B^3$** Different from MUC and CEAF, $B^3$ assigns a score to each mention instead of each entity. Here, we need a slightly different data structure, where we pair each mention with the entity it belongs to:

```
class Membership:   # An instance of a relation
  mention: Mention
  entity: Entity

class CorefOutputForB3:
  rels: Set[Membership]   # membership relations
```

The recall of $B^3$ assigns to each reference mention a score equal to the ratio of the number of correct mentions in the predicted entity containing the reference mention to the size of the reference entity to which that mention belongs (Pradhan et al., 2014). Under our new data structure this ratio is just $R^{\leftrightarrow}_{\text{entity}} [\delta]$. Precision is computed similarly by switching the role of predicted and reference entities. Thus, $B^3$ can be succinctly expressed as

$$R_{B^3} = R^{\leftrightarrow}_{\text{rels}} [\delta_{\text{mention}} \times R^{\leftrightarrow}_{\text{entity}} [\delta_{\text{mention}}]] \quad (23)$$

$$P_{B^3} = P^{\leftrightarrow}_{\text{rels}} [\delta_{\text{mention}} \times P^{\leftrightarrow}_{\text{entity}} [\delta_{\text{mention}}]]. \quad (24)$$

Our framework thus captures all three of the standard coreference resolution metrics.

### 3.3 Role-Filler Entity Extraction & *N*-ary Relation Extraction

Relation and event extraction generally take mentions as arguments, but some tasks take entities as arguments (*fillers*) of roles (*slots*) in some relation:

```
class RoleFillerEntity:
  role: RoleType
  entity: Entity

class NAryRelation:
  type: RelationType
  args: Set[RoleFillerEntity]

class NAryRelationSet:
  relations: Set[NAryRelation]
```

Tasks of this form have been instantiated in various ways in prior work, which we discuss below.

**Role-filler Entity Extraction** One such instance is *role-filler entity extraction* (REE), a subtask of template extraction in which one must populate the subset of slots (roles) of a *single* identified template that takes entities as fillers (Du et al., 2021a; Huang et al., 2021, *i.a.*). Since the task deals with a single template, the output is a single NAryRelation.[15]

Du et al. (2021a) introduced the CEAF-REE metric for REE which differs from CEAF only in requiring matching entities to share a role type and in using a different $\phi$ for entities:

$$\phi_\subseteq(P, R) := [\![ P \subseteq R ]\!] \qquad (25)$$

where $P$ and $R$ are predicted and reference entities (sets of mentions). CEAF-REE is then defined as:

$$\text{CEAF-REE} = \mathsf{F}^\leftrightarrow_{\text{args}} [\delta_{\text{role}} \times \phi_\subseteq] \qquad (26)$$

Whereas $\phi_3$ and $\phi_4$ award partial credit to predicted entities that contain at least one correct mention, $\phi_\subseteq$ is much stricter, awarding *no* credit in cases where even one mention is incorrect, while simultaneously awarding *full* credit to any non-empty subset of the reference entity. This may make sense in some settings, but in most, it is unduly harsh (see §6). Responding to this observation, Chen et al. (2023) suggest a pair of alternatives to CEAF-REE, CEAF-RME$_{\phi_\subseteq}$ and CEAF-RME$_{\phi_3}$, that treat predicted *mentions* as singleton entities and relax the

---

[15] Some work has evaluated at the mention level (Patwardhan and Riloff, 2009; Huang and Riloff, 2011; Du and Cardie, 2020), essentially doing named entity recognition (NER).

two-sided matching constraints to one-sided:

$$\text{CEAF-RME}_{\phi_\subseteq} = \mathsf{F}^\rightarrow_{\text{args}} [\delta_{\text{role}} \times \phi_\subseteq] \qquad (27)$$

$$\text{CEAF-RME}_{\phi_3} = \mathsf{F}^\rightarrow_{\text{args}} [\delta_{\text{role}} \times \phi_3] \qquad (28)$$

**N-ary Relation Extraction** *N*-ary RE generalizes binary RE to relations among *N entity* or *mention* arguments. Here, we will assume we are dealing with entities; the case of mention arguments is comparatively straightforward.

Often, work on *N*-ary RE assumes gold entities are given to the model as input, along with a set of candidate relations, and results are reported as relation type classification accuracy or $F_1$. This is true of much work on a number of recent, popular *N*-ary RE benchmarks, including SCIERC (Luan et al., 2018), DOCRED (Yao et al., 2019), and the dataset released by Peng et al. (2017).

In a more comprehensive task setting, entities or mentions must also be predicted, along with the relations. We highlight the SCIREX benchmark (Jain et al., 2020), an extension of SCIERC, as an example of evaluation in this setting. SCIREX requires extraction of quaternary (dataset, method, task, metric) relations over entities extracted from ML papers. We formulate the SCIREX metric in our framework below. For this task, mentions are represented as index ranges:

```
class Mention: # alternative to definition in §2.1
  indices: range   # set of token indices
```

A predicted mention is considered to match a reference mention iff their Jaccard similarity (considered as bag-of-integer offsets) exceeds 0.5:

$$\phi_{\text{Mention}} = [\![ \mathsf{J}^\leftrightarrow_{\text{indices}} [\delta_{\text{int}}] > 0.5 ]\!] \qquad (29)$$

Jain et al. propose computing a role-filling entity matching based on mention and role matching:

$$\phi_{\text{RFE}} = [\![ \mathsf{P}^\leftrightarrow_{\text{mentions}} [\delta_{\text{role}} \times \phi_{\text{Mention}}] > 0.5 ]\!]. \qquad (30)$$

In other words, a pair of entities $E_P$ and $E_R$ will be matched iff more than half of $E_P$'s mentions appear in $E_R$, and their role matches. Given this matching, predicted 4-ary relations are then evaluated against reference ones using $\mathsf{F}^\leftrightarrow[\phi_{\text{NAryRelation}}]$, where

$$\phi_{\text{NAryRelation}} = [\![ \mathsf{F}^\leftrightarrow_{\text{args}} [\phi_{\text{RFE}}] = 1 ]\!]. \qquad (31)$$

$\mathsf{F}^\leftrightarrow_{\text{args}} [\phi_{\text{RFE}}] = 1$ means that all four role-filler entities must match under $\phi_{\text{RFE}}$ to receive credit. $\mathsf{F}^\leftrightarrow_{\text{relations}} [\phi_{\text{NAryRelation}}]$ further illustrates how

matching superstructures depends on matching substructures, with optimal matching of relations depending on optimal matching of entities, which in turn depends on optimal matching of mentions.

### 3.4 Template Extraction

We now turn to *template extraction*, which arguably features the most complex outputs of any IE task. It generalizes $N$-ary RE by allowing roles in the relation be filled by any number of arguments $N \geq 0$, which may be of any type $\mathsf{T}$:

```
class SlotFiller[T]:
  slot: SlotType
  value: T  # Mention, Entity, Event, bool, etc.

class Template:
  type: TemplateType
  fillers: Set[SlotFiller[Any]]

class TemplateSet:  # task output
  templates: Set[Template]
```

where a distinct similarity function $\phi_\mathsf{T}$ may be needed for each $\mathsf{T}$. Constraints on template matchings are traditionally two-sided. Below, we consider the metrics employed for the classic MUC-4 task. In Appendix D, we also consider the more recent BETTER Granular benchmark.

The MUC-4 dataset (MUC, 1992; Sundheim, 1992) features 6 template types, which concern varieties of terrorist act (e.g. bombing, kidnapping) and which all contain the same slots. Some are "string-fill" slots, which take entity mentions as fillers, and others are "set-fill" slots, which take a categorical value. Although the official MUC-4 evaluation reported several metrics,[16] the *overall score* was $\mathrm{F_1}$ over slot fillers:

$$\mathsf{F}^{\leftrightarrow}_{\mathsf{templates}} \left[ \delta_{\mathsf{type}} \times \Sigma^{\leftrightarrow}_{\mathsf{fillers}} \left[ \delta_{\mathsf{slot}} \times \phi_\mathsf{T} \right] \right] \quad (32)$$

where $\phi_\mathsf{T} \in \{\phi_{\mathsf{set}}, \phi_{\mathsf{str}}\}$ is the type-appropriate filler similarity function. Both $\phi_{\mathsf{set}}$ and $\phi_{\mathsf{str}}$ are somewhat complex and, similar to $\phi_3$ and $\phi_4$, allow for partial credit. For some of the set-fill slots, the possible values are hierarchical; i.e., some values are more specific, and thus considered more accurate, than others. Suppose a set-fill slot $s$ takes values from a set $\mathcal{V}$, and we write $P <: R$ to denote $P$ is a subtype of $R$. Then $P <: R$ iff $P$ is a descendant of $R$ according to some hierarchy for

---

[16] See Chinchor (1992) for details.

$P, R \in \mathcal{V}$. MUC-4 defines $\phi_{\mathsf{set}}$ as:

$$\phi_{\mathsf{set}}(P, R) := \begin{cases} 1 & \text{if } P = R; \\ \frac{1}{2} & \text{if } P <: R; \\ 0 & \text{otherwise} \end{cases} \quad (33)$$

This choice of $\phi_{\mathsf{set}}$ is notable for suggesting a means of handling hierarchical sub-ontologies of the template ontology itself; such ontologies have seen considerable interest in many recent IE benchmarks, including RAMS (Ebner et al., 2020), WikiEvents (Li et al., 2021), and BETTER (Mckinnon and Rubino, 2022). We return to this in §6.

String-fill slots were evaluated based on maximum lexical overlap between a predicted mention and *all* mentions in a reference *entity*. We provide more detailed discussion in Appendix C.

## 4 Sets with Latent Variables

Next, we consider Abstract Meaning Representation (AMR) parsing (Langkilde and Knight, 1998; Banarescu et al., 2013), which involves outputs with *latent variables*. AMR describes the semantics of a sentence as a rooted, directed graph represented by a set of neo-Davidsonian triples, each with a subject, an object, and a relation. Subjects are variables and objects can be variables or concepts (e.g. from PropBank (Palmer et al., 2005)):

```
class Prop:  # logical propositions
  rel: Relation  # instance, ARG0, ARG1, ...
  subj: Var  # x, y, ...
  obj: Var | Concept  # z, want-01, boy, ...

class AMR:
  props: Set[Prop]
```

Following the metrics for relation extraction, a *prima facie* appealing metric for AMR graphs would be just like Eq. 10 for binary RE:

$$\mathsf{F}^{\leftrightarrow}_{\mathsf{props}}[\delta_{\mathsf{rel}} \times \phi_{\mathsf{subj}} \times \phi_{\mathsf{obj}}]$$

However, this poses a problem, as we cannot know whether two variables $x$ and $y$ refer to the same object: instance($x$, boy) and instance($y$, boy) could match if there is no constraint enforcing that $x \neq y$. Thus, it is not immediately clear what the similarity function for variables ($\phi_{\mathsf{Var}}$) should be.

The commonly used SMATCH metric solves this problem. SMATCH is defined to be the *maximum F-score obtainable via a one-to-one matching of variables between two AMRs* (Cai and Knight, 2013). That is, it looks for an optimal partial bijection

$M_V^{\leftrightarrow} \subseteq V_P \times V_R$ between the variables of the predicted and reference AMRs ($V_P$ and $V_R$, respectively). Given $M_V^{\leftrightarrow}$, we can define

$$\tilde{\phi}_{\mathsf{Var}}(x, y) = [\![(x, y) \in M_V^{\leftrightarrow}]\!], \qquad (34)$$

where $\tilde{\phi}$ denotes a similarity conditioned on the variables in its arguments being matched. Hence SMATCH is given by

$$\text{SMATCH} = \max_{M_V^{\leftrightarrow}} \mathsf{F}_{\mathsf{props}}^{\leftrightarrow}[\delta_{\mathsf{rel}} \times \tilde{\phi}_{\mathsf{subj}} \times \tilde{\phi}_{\mathsf{obj}}]. \ (35)$$

We generalize the spirit of SMATCH to any set of $X$ with *latent variables* yet to be matched. The *matching score* of $P, R$ with latent variables $V_P, V_R$ is defined to be

$$\Sigma^{\diamond}(P, R) = \max_{M_V^{\leftrightarrow}, M^{\diamond}} \sum_{(u,v) \in M} \tilde{\phi}_X(u, v), \qquad (36)$$

where $M_V^{\leftrightarrow}$ is an one-to-one matching between the variable set $V_P$ and $V_R$; and $M^{\diamond}$ is an matching between objects in $P$ and $R$ under constraint $\diamond$.

Computing this constrained optimization problem requires solving $M_V^{\leftrightarrow}$, which can be done via an integer linear programming (ILP) solver (Cai and Knight, 2013). See Appendix A for more details.

## 5 Matching of Other Structures

In the past few sections we developed tools to obtain *matching of sets*. We can extend this to match more complex structures such as *sequences*, *DAGs*, and arbitrary *directed graphs*.

Recall the matching score in Eq. 13: we computed a sum of similarities based on *matched pairs*. In the matching of structures, the matching should preserve the structure of the object being matched.

Elements of a *sequence* form a *total order* where earlier elements *precede* later elements. Given two sequences $P, R$ whose elements are of type $X$, each is equipped with a total order: $(P, \preceq_P), (R, \preceq_R)$. To compute the matching score of two sequences, we define

$$\Sigma^{\diamond}(P, R) = \max_{M^{\diamond}} \sum_{(u,v) \in M^{\diamond}} \phi_X(u, v) \qquad (37)$$

s.t. $\forall (u, v), (u', v') \in M^{\diamond}, u \preceq_P u' \iff v \preceq_R v'$.

That is, we seek a *maximum monotonic matching* between $P$ and $R$ that preserves the total order. For example, the matching score between $(1, 2, 3, 4, 5)$ and $(1, 3, 5, 7, 9)$ is 3 since $1, 3, 5$ are monotonically matched. The sequence matching

problem given by Eq. (37) is a weighted *longest common subsequence* (LCS) problem, and thus can be solved with dynamic programming.

We can further generalize this matching score to DAGs and graphs by noting that the total order $\preceq$ of sequence elements is relaxed to a *partial order* in DAGs and a *preorder* in arbitrary directed graphs. The constrained optimization problem in Eq. 37 can be solved via ILP, see Appendix A.

## 6 Discussion

We have seen that a diverse set of structured prediction metrics can be framed as computing a normalized total matching score for an optimal matching, given some similarity, which may *itself* reflect a score over an optimal matching of the relevant substructures. We now consider how different problem settings may motivate particular design decisions within this framework. We also highlight a couple of cases in which the actual metrics used for a task might be modified to better fit the problem setting.

**Partial Credit** For many tasks, we want to award some credit for partially correct responses. In applications where precision is paramount, it may be appropriate to insist on exact matches, but less so when some modest tradeoff with recall is desired. Moreover, many IE objects intuitively admit gradient notions of correctness.

Perhaps the most obvious example of this is mentions. Exact match on mentions, whether string- or offset-based, remains surprisingly common despite the possibility for variation in how they are annotated (e.g. disagreements about the extent of NPs). More relaxed mention similarities — such as head word matching or Jaccard score — are typically more appropriate. Recently, there has also been greater interest in the *informativity* of entity mentions (Li et al., 2021; Chen et al., 2023), where, e.g., names > nominal expressions > pronouns, and where scores may need to vary according to a mention's informativity. All of these can be captured by different choices of $\phi_{\mathsf{Mention}}$.

REE offers another example. Earlier (§3.3), we saw that CEAF-REE uses the $\phi_{\subseteq}$ entity similarity, which awards no credit at all to entities containing even one *incorrect* mention, but full credit to entities containing just one *correct* mention. A more natural extension of the CEAF metric to the REE setting, and one that permits partial credit, would be to replace $\phi_{\subseteq}$ with $\phi_3$ or $\phi_4$.

**Hierarchical Ontologies**    Type hierarchies are another common feature of IE problems: both events and entities may have types, subtypes, and even sub-subtypes. This is true of the event ontologies for FrameNet (Baker et al., 1998), RAMS (Ebner et al., 2020), WikiEvents (Li et al., 2021), and even MUC-4.[17]  Yet, the standard evaluation metrics for these datasets do not take the hierarchy into account, instead treating the ontology as flat.

Following the discussion above, it may thus often be appropriate to replace exact type matches ($\delta_{\text{type}}$) with similarities that award partial credit for correct ancestor type prediction. One possibility is a level-based partial scoring: Given a $D$-level type ontology with types specified as a $D$-tuple $P = (p_1, \ldots, p_D)$, we could, for instance, award credit based on the depth $d \in \{0, \ldots, D\}$ of the most specific correctly predicted type, e.g.:

$$\phi_{\text{type}}(P, R) = \begin{cases} 2^{d-D} & \text{if } d > 0; \\ 0 & \text{otherwise,} \end{cases} \quad (38)$$

where $d = 0$ iff even the most general type is incorrectly predicted. Or, one could adopt practices from related work in fine-grained entity typing (Ling and Weld, 2012; Chen et al., 2020, i.a.), which use the $F_1$ score of the set of all possible supertypes of predicted / reference types $S(P) = \{t | p <: t, p \in P\}$:

$$\phi_{\text{type}}(P, R) = F^{\leftrightarrow}[\delta_{\text{type}}](S(P), S(R)). \quad (39)$$

There is some precedent for schemes like this, but proper analysis of performance on tasks with hierarchical ontologies requires metrics that account for that hierarchy, and the field of IE as a whole would benefit from adopting them more widely.

**One-Sided vs. Two-Sided Constraints**    In general, metrics impose constraints on the matching between the predictions and the reference. Overwhelmingly, these tend to be two-sided (bijective) constraints, as systems usually try to generate just one predicted object for each one in the reference. But this is not always the case. The CEAF-RME metrics (Eqs. 27 and 28) proposed by Chen et al. (2023), which use one-sided constraints, are motivated in part by a need to evaluate a model that predicts *mention* fillers against references that contain *entity* fillers. This suggests a more general motivation for one-sided constraints — namely, for cases where the reference outputs are *sets*, but where predictions take the form of *members* of those sets.

---

[17] The `attack` template type is considered the parent type of all other template types in MUC-4.

# 7    Conclusion

We have presented a framework that unifies a variety of structured prediction metrics as normalized scores over (possibly hierarchical) constrained optimal matchings of structured objects. On the side of *theory*, our framework elucidates the relationships among tasks by defining the core components of their metrics. On the side of *practice*, it offers a compositional toolkit for the design of new metrics (aided by our library) and for critically evaluating existing ones, showing where they may inadequately capture important task features (§6). We intend this work to help the NLP community converge both on a common language for metric design and on more standardized metric implementations.

# Ethics Statement

As this work principally describes a conceptual framework and presents a survey of evaluation metrics, we do not believe it raises ethical concerns.

# Limitations

While this work aims to give a unified treatment of a variety of different metrics, our coverage of existing metrics is not exhaustive, and is intended rather to convey the expressiveness of the framework.

Our framework for evaluation is based on *matching* of substructures — thus metrics based on structure *editing* (e.g. string or tree edit distances; word error rate (WER) in speech recognition) cannot be expressed naturally in our formulation. One can of course define a $\phi$ based on edit distances over sequences, but that has to be an atomic definition and cannot be derived naturally under our bottom-up approach.

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

## A Solving ILPs

We use the integer linear programming (ILP) solver `scipy.optimize.milp` in SciPy (Virtanen et al., 2020), which wraps the HiGHS solver (Huangfu and Hall, 2018).

### A.1 Set Matching with Latent Variables

To solve the combinatorial optimization problem in Eq. (36)

$$\Sigma^\diamond(P, R) = \max_{M_V^\leftrightarrow, M^\diamond} \sum_{(u,v) \in M} \tilde{\phi}_X(u, v),$$

we cast it as an ILP problem. Recall that $P$ and $R$ contains variables in the set of $V_P, V_R$ respectively, and $\tilde{\phi}_X$ is a (unnormalized) similarity for $X$ *assuming* that their variables match. Essentially, $\tilde{\phi}_X(u, v)$ is an *upper bound* of the actual score the pair $(u, v)$ may obtain.

We create the following variables for ILP, where $\mathbb{B} = \{0, 1\}$:

- $m_{uv} \in \mathbb{B}$ is set to true if $u, v$ matches, i.e., $u \in P, v \in R$ and $(u, v) \in M^\diamond$.

- $\tilde{m}_{xy} \in \mathbb{B}$ is set to true if variables $x, y$ matches, i.e., $x \in V_P, y \in V_R$ and $(x, y) \in M_V^\leftrightarrow$.

We are solving the following ILP problem

$$\max \begin{bmatrix} \mathbf{c}^T & \mathbf{0}^T \end{bmatrix} \cdot \begin{bmatrix} \mathbf{m} \\ \tilde{\mathbf{m}} \end{bmatrix} \qquad (40)$$

with the vector to be solved being $\mathbf{m} = \text{vec}[(m_{uv})_{u \in P, v \in R}]$; $\tilde{\mathbf{m}} = \text{vec}[(\tilde{m}_{xy})_{x \in V_P, y \in V_R}]$, and the coefficient vector $\mathbf{c} = \text{vec}[(\tilde{\phi}_X(u, v))_{uv}]$ ("vec" is the matrix vectorization operator). The following constraints applied as needed.

**Latent Variable Constraints (Required)** Recall that $X$ is a product type and its fields are $\{\text{fld}_i : \text{type}_i\}$. If $\text{type}_i$ is $\text{Var}$ (the type for yet-to-be-matched variables), then $u$ matches $v$ *implies* that $u.\text{fld}_i$ matches $v.\text{fld}_i$. Translating this to constraints:

$$m_{uv} \le \tilde{m}_{u.\text{fld}_i, v.\text{fld}_i}, \quad \forall X.\text{fld}_i : \text{Var}. \qquad (41)$$

**1:1 Constraints on Variables (Required)** Each $x \in V_P$ can only be matched to at most one $y \in V_R$, and vice versa:

$$\sum_x \tilde{m}_{xy} \le 1; \quad \sum_y \tilde{m}_{xy} \le 1. \qquad (42)$$

**N:1 Constraints (Optional)** Each $u \in P$ can only be matched to at most one $v \in R$. This is naturally encoded as

$$\sum_v m_{uv} \le 1. \qquad (43)$$

**1:N Constraints (Optional)** Similarly, we have

$$\sum_u m_{uv} \le 1. \qquad (44)$$

**1:1 Constraints (Optional)** This is simply the two constraints above combined. This translation to ILP is a generalization of the method proposed in SMATCH (Cai and Knight, 2013).

### A.2 Matching of Arbitrary Structure

To solve the combinatorial optimization problem in Eq. (37)

$$\Sigma^\diamond(P, R) = \max_{M^\diamond} \sum_{(u,v) \in M^\diamond} \phi_X(u, v)$$

s.t. $\forall (u, v), (u', v') \in M^\diamond, u \preceq_P u' \iff v \preceq_R v',$

we cast it as an ILP problem similar to the translation above.

A variable $m_{uv} \in \mathbb{B}$ is set to true if $u \in P, v \in R$ and $(u, v) \in M$: i.e. $u$ and $v$ are matched.

We similarly set the coefficient vector $\mathbf{c}$ such that $c_{uv} = \phi_X(u, v)$. Therefore we are maximizing $\mathbf{c}^T \mathbf{m}$ under the monotonicity constraints

$$M^\diamond(u, v) \wedge M^\diamond(u', v') \Rightarrow u \preceq_P u' \Leftrightarrow v \preceq_R v'.$$

The monotonicity constraints can be encoded as linear constraints by the following:

$$m_{uv} + m_{u'v'} - 1 \le [\![u \preceq_P u' \Leftrightarrow v \preceq_R v']\!],$$

which can be rewritten as

$$m_{uv} + m_{u'v'} \le 1 + [\![u \preceq_P u' \Leftrightarrow v \preceq_R v']\!].$$

## B Kernels

Similarity functions have additional desirable properties when certain conditions are met. Below, we describe conditions under which some similarity functions are *(positive definite symmetric) kernels*.

**Definition 2.** *A **positive definite symmetric kernel** (p.d.s. kernel) is a function $\kappa : X \times X \to \mathbb{R}$ that satisfies symmetry $\kappa(x, y) = \kappa(y, x)$ and positive-semidefiniteness: $\mathbf{c}^T \mathbf{K} \mathbf{c} \ge 0$ where $\mathbf{K}_{ij} = \kappa(x_i, x_j)$, for all $x_1, \cdots, x_n \in X, \mathbf{c} \in \mathbb{R}^n$.*

**Lemma 1.** *The Kronecker $\delta : X \times X \rightarrow \{0, 1\}$ is a p.d.s. kernel.*

**Lemma 2.** ([Mohri et al., 2012](#), Theorem 6.10) *If $\kappa_1 : X_1 \times X_1 \rightarrow \mathbb{R}$ and $\kappa_2 : X_2 \times X_2 \rightarrow \mathbb{R}$ are p.d.s. kernels, then the product kernel $\kappa : (X_1 \times X_2) \times (X_1 \times X_2) \rightarrow \mathbb{R}$ is a p.d.s. kernel where $\kappa((x_1, x_2), (x_1', x_2')) = \kappa_1(x_1, x_1') \cdot \kappa_2(x_2, x_2')$.*

This is the kernel version of the *product similarity* we discussed in the main text.

**Lemma 3.** ([Haussler, 1999](#)) *If $\kappa$ is a kernel, the vertex label kernel $\Sigma^{\sim}[\kappa]$ is a kernel.*

**Definition 3.** ([Kriege et al., 2016](#)) *A kernel $\kappa : X \times X \rightarrow \mathbb{R}_{\geq 0}$ is a strong kernel if $\kappa(x, x) \geq \kappa(x, y)$ for all $x, y \in X$.*

**Lemma 4.** *A similarity function that is also a kernel is a strong kernel.*

**Lemma 5.** ([Kriege et al., 2016](#)) *If $\kappa$ is a strong kernel, the optimal assignment kernel $\Sigma^{\leftrightarrow}[\kappa]$ is a kernel.*

**Lemma 6.** *If $\kappa$ is a kernel bounded by above: $\max_{x,y} \kappa(x, y) < U < +\infty$, then $\kappa'(x, y) = \dfrac{1}{U - \kappa(x, y)}$ is a kernel.*

*Proof.* With Taylor expansion, we have

$$\kappa'(x, y) = \frac{1}{U} \sum_{i=0}^{\infty} \left( \frac{\kappa(x, y)}{U} \right)^i.$$

This series converges because $\kappa(x, y) < U$. Since kernels are closed under power series, this is a kernel. □

**Lemma 7.** *If $\kappa$ is a kernel, $\mathsf{F}[\kappa]$ is a kernel.*

*Proof.* Let $H(x, y) = \dfrac{1}{\kappa(x, x) + \kappa(y, y)}$. $H$ is a kernel. To see this, we prove that it is positive semidefinite. For all $n \in \mathbb{N}, c_i \in \mathbb{R}$, and $x_i \in X$ ($1 \leq i \leq n$), let $s_i = \kappa(x_i, x_i)$.

$$\sum_{i=1}^{n} \sum_{j=1}^{n} c_i \frac{1}{s_i + s_j} c_j$$
$$= \sum_{i=1}^{n} \sum_{j=1}^{n} c_i c_j \int_0^1 t^{s_i + s_j - 1} \mathrm{d}t$$
$$= \int_0^1 t \sum_{i=1}^{n} \sum_{j=1}^{n} c_i c_j t^{s_i - 1} t^{s_j - 1} \mathrm{d}t$$
$$= \int_0^1 t \left( \sum_{i=1}^{n} t^{s_i - 1} \right)^2 \mathrm{d}t \geq 0.$$

Hence $H$ is a kernel.

Note that F score can be written as

$$\mathsf{F}[\kappa](x, y) = \frac{2\kappa(x, y)}{\kappa(x, x) + \kappa(y, y)} = 2\kappa(x, y)H(x, y).$$

Hence $\mathsf{F}[\kappa]$ is a kernel. □

**Lemma 8.** *If $\kappa$ is a kernel, $\mathsf{J}[\kappa]$ is a kernel.*

*Proof.* Thus

$$\mathsf{J}[\kappa](x, y) = \frac{\kappa(x, y)}{\kappa(x, x) + \kappa(y, y) - \kappa(x, y)}$$
$$= \kappa(x, y) \frac{1}{1 - \kappa(x, y)H(x, y)}.$$

Since $2\kappa(x, y)H(x, y) = \mathsf{F}[\kappa](x, y) \leq 1$, we have $\kappa(x, y)H(x, y) \leq \dfrac{1}{2} < 1$. By Lemma 6, $\dfrac{1}{1 - \kappa(x, y)H(x, y)}$ is a kernel, so $\mathsf{J}[\kappa]$ is a kernel. □

Therefore, given the lemmata above, we have the nice property that any kernels composed with $\mathsf{F}^{\leftrightarrow}$, $\mathsf{J}^{\leftrightarrow}$ are kernels. The deductions presented here follows [Shen (2019)](#).

## C MUC-4 Evaluation: Additional Details

**String-valued similarities and Interactive Scoring** For string-valued slots, although full entities are annotated in the reference, systems are required to predict just one mention per entity. Two different versions of $\phi_{\mathsf{str}}$ were used: one for determining the template alignment and one for computing the score given that alignment. The first version awarded full credit when there was at least a one-word overlap between the predicted string and at least *one* of the strings in the reference entity, so long as that word was not a designated premodifier; zero credit was awarded otherwise. Suppose $\mathtt{valid}(P, R)$ is true iff *neither* word $P$ nor word $R$ is a premodifier. Then we can write:

$$\phi_{\mathsf{word}}(P, R) = \delta_{\mathsf{word}} \cdot [\![\mathtt{valid}(P, R)]\!] \tag{45}$$
$$\phi_{\mathsf{str}} = [\![\mathsf{J}_{\mathsf{words}}^{\leftrightarrow}[\phi_{\mathsf{word}}] > 0]\!] \tag{46}$$

The second version of $\phi_{\mathsf{str}}$, used for final reporting, merely enhanced Eq. 46 by interactively querying the user in cases where a mismatch could not be automatically resolved, whereupon the user could determine the appropriate amount of credit to award, including partial (=half) credit. Full guidelines on interactive scoring can be found in the gzip archive containing the MUC-3 and MUC-4 data here: `https://www-nlpir.nist.gov/related_projects/muc/muc_data/muc_data_index.html` (see `TEST/SCORER/scoring-guidelines.v7`).

**Template Alignment Constraints** Template alignments for the original evaluation featured a couple of quirks. For one, it was possible to obtain partial (=half) credit for the template ("incident") type by predicting the generic `attack` label in place of any of the other, more specific labels (`bombing`, `kidnapping`, etc.).[18] For another, a partial match on at least one of the following slots was required: physical target identifier, physical target type, human target name, human target description, human target type, perpetrator individual identifier, and perpetrator organization identifier. Chinchor (1992) notes that this constraint was put in place to prevent "fortuitous" but spurious template alignments that were observed in the MUC-3 evaluation. To our knowledge, researchers have not applied these rules in evaluating their own systems on MUC-4 since the original evaluation.

**MUC-4: Recent Work** In recent years, it has become standard to evaluate only on the string-fill slots, plus the template type (Chambers and Jurafsky, 2011; Du et al., 2021b; Das et al., 2022; Chen et al., 2023). Du et al. (2021b) thus proposed a version of Eq. 32 that sets $\phi_T = \phi_\subseteq$, which amounts to using CEAF-REE (Eq. 26) to determine the optimal template alignment. Following on this work, Chen et al. (2023) additionally present MUC-4 results using their relaxed (one-sided) metrics (Eqs. 27, 28) for $\phi_T$.

## D   BETTER

BETTER Granular is a recent template extraction dataset released as part of the IARPA BETTER program that is more complex than MUC-4 both in having different slots for each template type and in having a greater diversity of filler types. Here, we focus just on the key difference in overall score calculation compared to MUC-4. The Granular score is the *product* of the slot filler $F_1$ score (Eq. 32) and the template type $F_1$ score:

$$
\begin{aligned}
&F^{\leftrightarrow}_{\texttt{templates}}[\delta_{\texttt{type}}] \\
&\times F^{\leftrightarrow}_{\texttt{templates}}\left[\delta_{\texttt{type}} \times \Sigma^{\leftrightarrow}_{\texttt{fillers}}\left[\delta_{\texttt{slot}} \times \phi_T\right]\right] \quad (47)
\end{aligned}
$$

where $\delta_{\texttt{type}}$ applies to template types and $\phi_T$ applies to filler types, as in Eq. 32. Because this product cannot be expressed as a sum of scores over aligned template pairs (Eq. 13), it does not, on its face, fit within our framework. However, this score could

still be optimized *indirectly* by instead optimizing the template alignment against the second term only, as this will be non-zero only in cases where there is a match on template type.

For more on BETTER, see Soboroff (2023), Mckinnon and Rubino (2022), and the following URL: `https://www.iarpa.gov/index.php/research-programs/better`. All BETTER program datasets are available here (note that account registration is required, but is free): `https://ir.nist.gov/better/`. Appendices C and D of Chen et al. (2023) also provide a good overview of the Granular task and its evaluation, including definitions of $\phi_T$ for all slot filler types.[19]

---

[18] Note that, as with scoring for set-fill slots, this presages the proposal in §6 for hierarchy-aware scoring.

[19] One distinctive feature of BETTER Granular in contrast to MUC-4 is that some slots may take events (as in §3.1) as fillers, in addition to entities and categorical values.