# OpenReview forum: "A Unified View of Evaluation Metrics for Structured Prediction"
_EMNLP/2023/Conference — EMNLP 2023 Main_

### Official Review · Reviewer_z6Hc · 2023-07-26

**Soundness:** 4

**Excitement:**

3: Ambivalent: It has merits (e.g., it reports state-of-the-art results, the idea is nice), but there are key weaknesses (e.g., it describes incremental work), and it can significantly benefit from another round of revision. However, I won't object to accepting it if my co-reviewers champion it.

**Paper Topic And Main Contributions:**

The authors present a theoretic framework of evaluation metrics for IE tasks, including binary/n-ary relation extraction, event extraction, coreference resolution, role-filler entity extraction and template extraction. This framework can server as a common language for evaluation metrics across the various IE tasks.

**Reasons To Accept:**

First theoretic framework of evaluation metrics for various IE tasks

**Reasons To Reject:**

- Lack of example applications of the framework for some widely used specific metrics of IE evaluation
- Lack of explanation of limitations of the framework, if it can be applied e.g. for the task of extracting tuples of various sizes, for open domain IE, where event types are not finite, for generative models, instead of extractive models, if the framework favors an IE system that produces more tuples than another producing less tuples, if the two constraints on the alignment in Section 2 are too restrictive
- Lack of some details, e.g. how to establish bijection, if the framework works at the sentence and/or document level

**Reproducibility:**

4: Could mostly reproduce the results, but there may be some variation because of sample variance or minor variations in their interpretation of the protocol or method.

**Reviewer Confidence:**

3: Pretty sure, but there's a chance I missed something. Although I have a good feel for this area in general, I did not carefully check the paper's details, e.g., the math, experimental design, or novelty.

---

> ### Author Rebuttal · Authors · 2023-08-29
>
> We sincerely appreciate this review and are pleased that you see merit in the framework! We hope to address your remaining concerns below.
>
>  > *Lack of example applications of the framework for some widely used specific metrics of IE evaluation*
>
> While we acknowledge that our survey of IE metrics is certainly not exhaustive (L189-92) — indeed, it could not be, given the space — we do try to present a broad and representative collection of actual, widely used metrics across multiple tasks and show how they realize our framework. Additionally, we use the framework to propose some alternative metric designs in section 4. However, we think it's possible we've not fully understood the concern raised here. For instance, perhaps there are specific metrics you have in mind that we do not cover that you believe we really ought to. We would appreciate further elaboration on this point.
>
> > *Lack of explanation of limitations of the framework …*
>
>  Thank you for highlighting these additional considerations. We do note two important limitations in the **Limitations** section — namely, that we cannot claim exhaustive coverage of all metrics and that there may be some metrics (e.g. MUC) that do fit the framework *(though see our response to Reviewer J1er on this point as we have expanded the framework to incorporate MUC and B3)*. We will be sure to address your concerns in our revisions, but below are some brief replies. In general, we think they represent points we could clarify, rather than genuine limitations.
>
>  - *Extraction of variably sized tuples*: This can readily be accommodated, so long as a similarity function can be defined between tuples of different sizes (though plausibly, one would want to set similarity between tuples of different arity to zero).
>  - *Infinite types (open-domain IE)*: Infinite label spaces can also be accommodated, once again so long as a similarity function can be defined between the relevant objects. Exact label match, for instance, does not require having a finite label set.
>  - *Generative vs. extractive models*: The framework deals with data structures that are generally specified by the task, not the model. As such, the framework is agnostic to model choice, so long as models produce outputs that can be mapped to these data structures (which is presumably true of any model designed for the task).
>  - *Favoring more tuples over fewer tuples*: The framework itself exhibits no preference in this regard. Granted, it is certainly possible to design a metric using the framework that does have such a preference for more tuples (e.g. any recall-normalized metric will have this property). But it is also possible to design a metric with the opposite preference (e.g. any precision-normalized metric).
>  - *Restrictive alignment constraints*: These constraints are fundamentally just descriptive of what we actually observe across IE metrics: in the vast majority of cases, each predicted object is paired with a single reference object and vice-versa (two-sided one-to-one constraints). Much more rarely, multiple predictions may correspond to the same reference (as for CEAF-RME; one-sided constraints). As we have responded to Reviewer J1er, we included a relaxed alignment with *no constraints*: this would let our framework encompass the MUC metric for coreference resolution.
>
> > *Lack of some details, e.g. how to establish bijection, if the framework works at the sentence and/or document level*
>
> Thank you for pointing out places where we could make our exposition clearer; we will happily include further clarifications on these points. To briefly address them here:
>
>  - (Partial) bijections are established via optimization over the metric itself: selecting the mapping between predicted and reference objects that maximizes total metric score (e.g., under one-to-one alignment constraints, it can be solved with the Hungarian algorithm). This optimization problem is the one at the core of all *unbalanced assignment problems* (L126-132) but we will make this more explicit.
>  - The framework is **fully agnostic to choice of text segmentation** (i.e. it can be applied to both sentence- and document-level problems).
>
> If there are further details we can clarify, please let us know.

---

### Official Review · Reviewer_Aomu · 2023-08-02

**Soundness:** 4

**Excitement:**

4: Strong: This paper deepens the understanding of some phenomenon or lowers the barriers to an existing research direction.

**Missing References:**

Not really a missing reference, but an interesting parallel work to check

https://aclanthology.org/2022.acl-long.395.pdf


**Paper Topic And Main Contributions:**

In this paper, the authors present two main contributions in the field of Information Extraction (IE).
First, they propose a framework to enhance the IE theory by highlighting the interconnections among different evaluation metrics proposed over the years. Second, they provide a systematic bottom-up approach for devising new metrics.

**Questions For The Authors:**

I wonder if there is any possibility to formally test with this framework whether a new evaluation measure is inconsistent or not robust.



**Reasons To Accept:**

The theoretical aspects are well presented and the framework will be of interest not only for the IE field.
The same approach (a generalization of evaluation measures) could be applied (provided the necessary adaptations of the problems that do not take into account the same marching constraints) to other fields such as information retrieval and text classification, for example.



**Reasons To Reject:**

The only issue that I can see is that, despite being a clear paper, the authors have added an overload of symbols for the mathematical notation.
Off the top of my head, I do not have a solution for this. I can only imagine that in order to have a unified view, this solution is the easiest one.

As a minor issue, the examples of the data structures are good at the beginning of the paper. After a couple of examples, they are redundant.

**Reproducibility:**

N/A: Doesn't apply, since the paper does not include empirical results.

**Reviewer Confidence:**

3: Pretty sure, but there's a chance I missed something. Although I have a good feel for this area in general, I did not carefully check the paper's details, e.g., the math, experimental design, or novelty.

---

> ### Author Rebuttal · Authors · 2023-08-29
>
> We thank you for this encouraging review!
>
> #### **Reasons to Reject**
>
>  - *On mathematical notations:* Since we unified various metrics in terms of their *substructures*,  *alignment constraints*, and *normalizers*, we believe that our notation is needed to fully express these features of various metrics. We ourselves deliberated quite a bit over the clearest notation to use and would be open to any specific suggestions for improving it.
>  - *On data structures:* We also acknowledge there is a risk of redundancy in giving explicit data structures for each problem we cover — particularly for readers intimately familiar with those problems. Our aim here was simply to avoid ambiguities **as well as to make the paper self-contained**, as similar problem settings often focus on different kinds of objects across benchmarks (e.g. extraction of mentions vs. entities in template filling and $N$-ary relation extraction). But we would likewise welcome recommendations for a more economical way of achieving this aim.
>
> #### **Questions**
>
> > *I wonder if there is any possibility to formally test with this framework whether a new evaluation measure is inconsistent or not robust.*
>
> This is an intriguing thought. However, we don't view the framework as adjudicating questions of robustness or inconsistency *per se*. As we suggest at different points (principally in section 4), the framework clearly admits metrics that have less than desirable properties (e.g. inability to award partial credit, insensitivity to an ontology's type hierarchy), while also admitting many standard, widely accepted metrics (trigger and argument F1 for event extraction). As such, we wouldn't contend that the framework cuts "robustness" or "inconsistency" at the joints, but it *does* isolate the key metric components that one might want to evaluate with respect to these characteristics for whatever problem is at hand.
>
> #### **Missing References**
>
> We appreciate the pointer to Lu et al. (2022), which is clearly relevant. We will be sure to include this citation in our revisions.

---

### Official Review · Reviewer_J1er · 2023-08-02

**Soundness:** 4

**Excitement:**

4: Strong: This paper deepens the understanding of some phenomenon or lowers the barriers to an existing research direction.

**Paper Topic And Main Contributions:**

This paper provides a general formalism for IE extraction, defining three components: similarity function, alignment, and normalizing function, and show how many IE tasks can be described that way: from binary relation extraction to MUC4 template extraction. The paper also applies the framework to other non-IE tasks, and suggests that its use can improve the design of new metrics.


**Questions For The Authors:**

Question A: Is there a convincing explanation/justification for your framework, independent of actual practice? That is, why should all IE-evaluation measures be based on your formulation, when there are actual measures which cannot be conveyed that way? To me, the (real) problems of MUC and B3 are not a reason why they should be discarded. Would it be possible to enlarge your framework so that they could be described there?

**Reasons To Accept:**

This paper provides an interesting generalization over many different evaluation measures in information extraction.

**Reasons To Reject:**

Since not all IE-evaluation measures can be translated into this framework, maybe it is not that relevant.

**Reproducibility:**

N/A: Doesn't apply, since the paper does not include empirical results.

**Reviewer Confidence:**

5: Positive that my evaluation is correct. I read the paper very carefully and I am very familiar with related work.

---

> ### Author Rebuttal · Authors · 2023-08-29
>
> We are grateful for this positive review and hope to speak to your concerns below.
>
> Upon deeper reflection, we have realized that our framework actually does encompass both precision and recall for MUC and $\rm B^3$, and we intend to update the appendix to reflect this. We should clarify that we were never intending to dismiss these metrics on the basis of their problems.
>
>  To encompass MUC, we first need to extend our notion of alignment constraints. Right now in our formulation there are two kinds of constraints: a one-to-one mapping, and a many-to-one mapping. We introduce a many-to-many mapping here, essentially with no constraints: the score would just be the sum of all edges in the bipartite graph. Let’s denote this case as $\sim$ (as opposed to $\to$, since we are moving from a *function* to any binary *relation*).
>
>    - MUC considers the number of *coreferent links* in entities. Between two entities $X$ and $Y$, the number of shared links between them is $\left| X \cup Y \right| - 1$ if they have more than 2 common mentions. Define this (shared number of coreferent links) as a score between a predicted entity and a reference entity: $ \phi_{\rm link}(X, Y) = \max (0, \left| X \cup Y \right| - 1 ) $.
>   Given the many-to-many mapping we just introduced (since MUC computes this score for *each pair* of predicted and reference entities), the precision of MUC would just be
>   $$ \mathtt{P}\_{\rm MUC} = \mathtt{P}\_\texttt{entities}^{\sim} [\phi_{\rm link}] $$
>   Recall/F1 can be defined similarly.
>  - Since $\rm B^3$ assigns a score to each *mention* based on the *entity it belongs to*, we need to slightly modify the data structure of the output of coreference resolution system to encompass $\rm B^3$: the type to derive a metric changes from *entities* to *(mention, entity) pairs*, just like a relation triplet describing a *membership* relation below. Under this output data structure, we can derive the precision and recall of $\rm B^3$ on the `MembershipRelationSet` type as:
> $$ \mathtt{P}\_{\rm B^3} = \mathtt{P}\_\texttt{rels}^{\leftrightarrow} [\delta\_\texttt{mention} \times \mathtt{P}\_\texttt{entity}^{\leftrightarrow}[\delta\_\texttt{mention}]] $$
> $$ \mathtt{R}\_{\rm B^3} = \mathtt{R}\_\texttt{rels}^{\leftrightarrow} [\delta\_\texttt{mention} \times \mathtt{R}\_\texttt{entity}^{\leftrightarrow}[\delta\_\texttt{mention}]] $$
>
> ```
>   class Membership:
>        mention: Mention
>        entity: Entity
>   class MembershipRelationSet:
>        rels: Set[Membership]
> ```
>
> ---
>
> > *Is there a convincing explanation/justification for your framework, independent of actual practice? That is, why should all IE-evaluation measures be based on your formulation, when there are actual measures which cannot be conveyed that way?*
>
> Even given our inclusion of MUC and $B^3$, we want to engage this point; it’s a good one. Suppose there *is* some other IE metric that in fact does not fit within the framework. We think the framework still has significant value for at least a couple of reasons.
>
>  - First, it clearly is descriptive of a broad set of important IE metrics, and even generalizations that admit of exceptions can still be tremendously helpful by illuminating key commonalities across a range of superficially disparate things. We are *not* making a prescriptive argument that all IE metrics absolutely *must* fit this framework. Rather, we’re making a descriptive argument that it is intriguing and informative that so many of them do.
>
>  - Second, as we contend in the introduction (L058-L060) and try to demonstrate in section 4, we think the framework is plausibly *generative*, inasmuch as it facilitates metric design by encouraging researchers to consider how the characteristics of the problem they’re working on are best reflected in the three framework components — constraints, similarity function, and normalization — and in how the scores for substructures compose in the scores for superstructures.
>
> Thus, we take the justification to consist not only in “actual practice” (i.e. the descriptive characterization of existing metrics), but also in its generative potential. We think these are sufficient. If there is another form of justification that seems important that we have not spoken to, we would be keen to hear of it.

---

### Meta-Review · Area_Chair_PXNH · 2023-09-18

**Recommendation:** 4

**Metareview:**

The paper provides a general formalism for several Information Extraction tasks, ranging from binary relation extraction to MUC4 template extraction, utilizing similarity functions, alignment, and normalization functions.

Overall, the paper is well-written and offers interesting generalizations across various evaluation measures in Information Extraction. It is also the first theoretically motivated framework in this field.

---

### Decision · Program_Chairs · 2023-10-07

**Decision:**

Accept-Main

**Comment:**

The paper provides a general formalism for several Information Extraction tasks, ranging from binary relation extraction to MUC4 template extraction, utilizing similarity functions, alignment, and normalization functions.

Overall, the paper is well-written and offers interesting generalizations across various evaluation measures in Information Extraction. It is also the first theoretically motivated framework in this field.